Noyes et al. Implementation Science 2018, **13**(Suppl 1):4

Implementation Science

**METHOD**

# Applying GRADE-CERQual to qualitative evidence synthesis findings–paper 6: how to assess relevance of the data

Jane Noyes[1], Andrew Booth[2*], Simon Lewin[3,4], Benedicte Carlsen[5], Claire Glenton[3], Christopher J. Colvin[6], Ruth Garside[7], Meghan A. Bohren[8], Arash Rashidian[9,10], Megan Wainwright[5], Özge Tunçalp [7], Jacqueline Chandler[11], Signe Flottorp[3], Tomas Pantoja[12], Joseph D. Tucker[13] and Heather Munthe-Kaas[3]

## Abstract

**Background:** The GRADE-CERQual (Confidence in Evidence from Reviews of Qualitative research) approach has been developed by the GRADE (Grading of Recommendations Assessment, Development and Evaluation) Working Group. The approach has been developed to support the use of findings from qualitative evidence syntheses in decision-making, including guideline development and policy formulation.

CERQual includes four components for assessing how much confidence to place in findings from reviews of qualitative research (also referred to as qualitative evidence syntheses): (1) methodological limitations, (2) coherence, (3) adequacy of data and (4) relevance. This paper is part of a series providing guidance on how to apply CERQual and focuses on CERQual's relevance component.

**Methods:** We developed the relevance component by searching the literature for definitions, gathering feedback from relevant research communities and developing consensus through project group meetings. We tested the CERQual relevance component within several qualitative evidence syntheses before agreeing on the current definition and principles for application.

**Results:** When applying CERQual, we define relevance as the extent to which the body of data from the primary studies supporting a review finding is applicable to the context (perspective or population, phenomenon of interest, setting) specified in the review question. In this paper, we describe the relevance component and its rationale and offer guidance on how to assess relevance in the context of a review finding. This guidance outlines the information required to assess relevance, the steps that need to be taken to assess relevance and examples of relevance assessments.

**Conclusions:** This paper provides guidance for review authors and others on undertaking an assessment of relevance in the context of the CERQual approach. Assessing the relevance component requires consideration of potentially important contextual factors at an early stage in the review process. We expect the CERQual approach, and its individual components, to develop further as our experiences with the practical implementation of the approach increase.

**Keywords:** Qualitative research, Qualitative evidence synthesis, Systematic review methodology, Research design, Methodology, Confidence, Guidance, Evidence-based practice, Relevance, GRADE

\* Correspondence: A.Booth@sheffield.ac.uk
[2]School of Health and Related Research (ScHARR), University of Sheffield, Sheffield, UK
Full list of author information is available at the end of the article

## Background

The GRADE-CERQual (Confidence in Evidence from Reviews of Qualitative research) approach has been developed by the GRADE (Grading of Recommendations Assessment, Development and Evaluation) Working Group. The approach has been developed to support the use of findings from qualitative evidence syntheses in decision-making, including guideline development and policy formulation. GRADE-CERQual (hereafter referred to as CERQual) includes four components for assessing how much confidence to place in findings from reviews of qualitative research (also referred to as qualitative evidence syntheses): (1) methodological limitations, (2) coherence, (3) adequacy of data and (4) relevance. This paper focuses on one of these four components, relevance.

When carrying out a CERQual assessment, we define relevance as the extent to which the body of data from the primary studies supporting a review finding is applicable to the context (including for example the perspective or population, phenomenon of interest, setting) specified in the review question [1]. The relevance component in CERQual is analogous to the indirectness domain used in the GRADE approach for assessing confidence in findings from systematic reviews of effectiveness [2].

## Aim

The aim of this paper, part of a series (Fig. 1), is to describe what we mean by relevance of data in the context of a qualitative evidence synthesis and to give guidance on how to operationalise this component in the context of a review finding as part of the CERQual approach. This paper should be read in conjunction with the papers describing the other three CERQual components [3–5] and the paper describing how to make an overall CERQual assessment of confidence and create a Summary of Qualitative Findings table [6]. Key definitions for the series are provided in Additional file 1.

## How CERQual was developed

The initial stages of the process for developing CERQual, which started in 2010, are outlined elsewhere [1]. Since then, we have further refined the current definitions of each component and the principles for application of the overall approach using a number of methods. When developing CERQual's relevance component, we undertook informal searches of the literature, including Google and Google Scholar, for definitions and discussion papers related to the concept of relevance and to related concepts such as internal and external validity. We carried out similar searches for the other three components. We

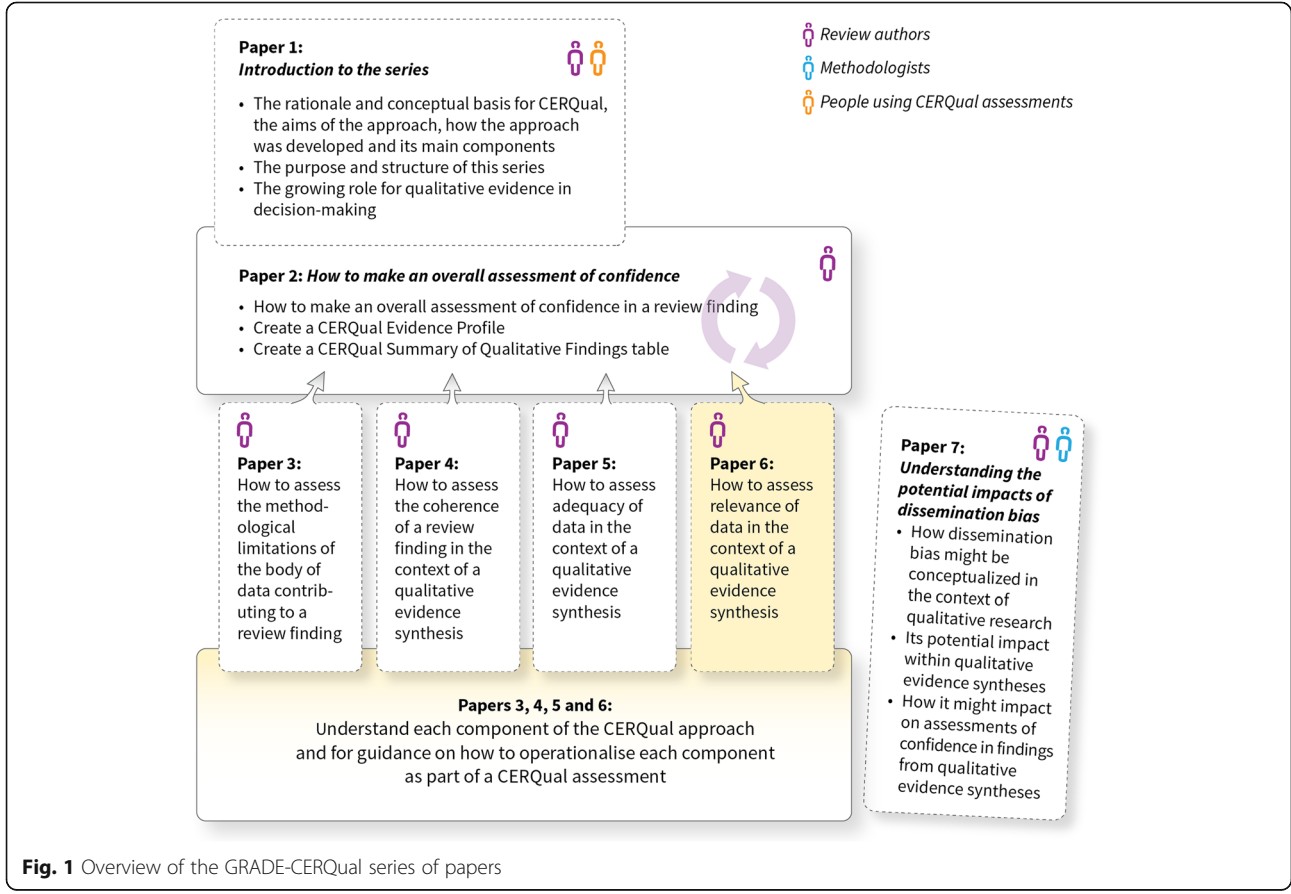

**Fig. 1** Overview of the GRADE-CERQual series of papers

presented an early version of the CERQual approach in 2015 to a group of methodologists, researchers and end users with experience in qualitative research, GRADE or guideline development. We further refined the approach through training workshops, seminars and presentations during which we actively sought, collated and shared feedback; by facilitating discussions of individual CERQual components within relevant organisations; through applying the approach within diverse qualitative evidence syntheses [7–17]; and through supporting other teams in using CERQual [18, 19]. As far as possible, we used a consensus approach in these processes. We also gathered feedback from CERQual users through an online feedback form and through short individual discussions. The methods used to develop CERQual are described in more detail in the first paper in this series [20].

### Assessing relevance

As described above, relevance in the context of CERQual is defined as the extent to which the body of data from the primary studies supporting a review finding is applicable to the context specified in the review question [1]. Relevance is the CERQual component that is anchored to the context specified in the review question. How the review question and objectives are expressed, how a priori subgroup analyses are specified and how theoretical considerations inform the review design are therefore critical to making an assessment of relevance when applying CERQual.

This paper primarily targets review authors who apply CERQual whilst undertaking their own review. We recognise additional circumstances when it is desirable to apply CERQual retrospectively to a review conducted by others. We envisage two scenarios when applying CERQual retrospectively (i) using the original review question or (ii) adapting the context specified in the original review question to reflect a particular context of interest (for example, an a posteriori identified 'subgroup' population or intervention type). These circumstances will feature in subsequent guidance, following additional methodological development and testing. For the remainder of this paper, we focus primarily on review authors who apply CERQual when undertaking their own review. (See also Additional file 2 in paper 2 in this series [6] for more information about applying CERQual to a synthesis conducted by others.)

### Description of and rationale for the relevance component

Evidence-informed health policymaking requires "relevant research" [21] that relates to the review question, which may be defined from initiation of the review or may emerge iteratively during the review process, and to the type of question that is being asked. The

methodological literature similarly refers to the degree of fit of included studies to the structured review question as one type of internal validity [22]. Internal validity, a term commonly used by systematic review authors, therefore maps onto the 'internal relevance' of included studies to the review question in a qualitative evidence synthesis.

In most qualitative evidence syntheses, the inclusion criteria for studies closely mirror the review question. Included studies must, by definition, be directly relevant to the review question. In other cases, the body of data from the primary studies may not be completely relevant because of differences between these data and the context of the review question, for instance differences in the focus of the study question, perspective or population, the phenomenon of interest or intervention, the setting or the timeframe. In some cases, the body of data from the included primary studies only partially address the context specified by the review question. This occurs if the body of data from the included studies covers only a subgroup of the population or a subtype of the intervention. In other cases, relevant studies are missing entirely. Under these circumstances, review authors sometimes extend the scope of eligible studies to include studies linked to the review question conceptually or by analogy.

CERQual focuses on the assessment of the *internal* relevance of the body of evidence from included studies contributing to a review finding, as mapped against the context of the review question. This assessment of relevance is not intended to make externally referent claims regarding the transferability, generalisability or applicability (terms that we take to mean the same) of a review finding. Wider *external* relevance (a concept that maps onto external validity––a term also commonly used by review authors conducting quantitative systematic reviews of the effectiveness of interventions) of a review finding is addressed in part by the overall CERQual assessment. This overall assessment, based on judgements for all four CERQual components, seeks to establish the extent to which a synthesis finding is a reasonable representation of the phenomenon of interest [1, 6]. An overall CERQual assessment communicates the extent to which the synthesis finding is likely to be substantially different from the phenomenon of interest, as defined in the review question. For completeness, an explanation of external relevance can be found in Additional file 2.

When assessing relevance in the context of CERQual, our aim is *not* to judge whether some absolute standard of relevance has been *achieved*, but to judge whether there are *grounds for concern* regarding relevance. Having identified any such concerns, we must consider whether these are sufficiently serious to lower our confidence in the review finding. We are likely to have

concerns about relevance when the context of the included studies does not adequately reflect the context determined by the review question.

### Guidance on how to assess relevance of data in the context of a review finding (Fig. 2)

#### Step 1: clarify the review question and context

When designing any review and developing the protocol, you should consider and decide which contextual factors are important for the review question [23]. By 'context', we refer to a complex and interacting composite that includes, but is not restricted to, the perspective, the population, the phenomenon of interest and the setting (Table 1). Review questions can be articulated using frameworks such as SPICE (Setting Perspective, Phenomenon of Interest, Evaluation) or adapted versions of PICO (Population, Phenomenon of Interest, Comparison, Outcomes) [24].

A priori identification of factors considered to be relevant to an individual review may also be facilitated, before commencement of a review, by reference to an available context framework or theory. Alternatively, a decision on likely factors may be informed by an existing review in a similar topic area. Incorporating insights from external frameworks, theories or comparable reviews at the design stage may help with a later assessment of relevance. For example, a specific theoretical

**Fig. 2** Steps when assessing methodological limitations

**Table 1** Contextual factors to consider when refining the review scope and specifying the question

Specify the context of the review question, including:

Micro-context

*The population*––specify any specific characteristics, perspectives or subgroups of the population (e.g. pregnant African women living in African countries)

*The setting*––such as hospital, private provider, timeframe of interest (e.g. publicly funded hospitals from 2000 to present time)

*The place*––such as geographical location, political system (e.g. African countries––state-funded healthcare)

Meso-context

*The intervention (where applicable)*––specify the intervention and components of interest (e.g. medically assisted birth in a state-funded hospital)

Macro-context

*The policy, political issues, social climate or legislation*––such as the policy context and legal framework associated with the phenomenon of interest (e.g. updating clinical and midwifery guidelines from African countries to promote safer birth and woman-centred care)

Cross cutting

*The phenomenon of interest*––(e.g. the experiences of African women regarding medically assisted birth in public hospitals in African countries)

Suggested frameworks for considering context include PROGRESS-Plus [33], the PRISMA-Equity extension [34] and the CICI Framework [35]. PROGRESS is an acronym for Place of Residence, Race/ Ethnicity, Occupation, Gender, Religion, Education, Socioeconomic Status, and Social Capital and Plus represents additional categories such as Age, Disability and Sexual Orientation

Suggested frameworks for describing the intervention include the i_CAT_SR tool [36] and TIDIER [37].

perspective (such as a behavioural theory [25]) or lens (such as an equity lens [26]) may be adopted to help define and refine the review question and important contextual factors. Identification, selection and application of an appropriate theory are critical to this process, and Cochrane Guidance is available for review authors to aid in selecting social theories in reviews [27].

Depending on the methodology selected, you may specify the review question(s) a priori in the protocol or may develop question(s) iteratively as part of the review process, with the final question(s) being documented later. Information about the context of the review question may appear in several sections of the review protocol, including the structured review question, inclusion/ exclusion criteria and any parameters set for the search and retrieval of studies [28]. The review's inclusion and exclusion criteria are thus an important additional source of information when assessing relevance [28]. Typically, these criteria include the geographical and temporal setting and the characteristics of the population, such as gender, ethnicity, religion and other demographic and cultural factors––the micro-context for the review (Table 1).

If the review explores how an intervention is implemented, then the review's inclusion and exclusion criteria are also likely to include the characteristics of the intervention and of those administering the intervention, and possibly also the organisational characteristics within which the intervention is delivered (the meso-context). For other review questions, particularly those related to policy, the political and legislative background context (the macro-context) are equally, if not more, important.

You may also augment the core review question, for instance by pre-specifying specific subgroups of the population or setting (for instance, young men or rural health facilities) that you will consider separately in the analysis. Pre-specification of subgroup analyses offers review authors the opportunity to apply CERQual to subgroup level findings. However, we recognise that review authors may only uncover meaningful groupings as they conduct the review itself.

If you are applying CERQual to someone else's review and the original question and context remain unchanged, then step 1 would apply in the same way. At present, we have no empirical base for guidance on applying CERQual to someone else's review when aspects of the original context of the review question have been changed to reflect a new context. We have yet to undertake the methodological development and testing for this specific application of CERQual.

### Step 2: decide on the appropriateness and implications of the study inclusion strategy
As a review author, you need to make an informed judgement on how and why you selected primary studies in relation to your review question. In a qualitative evidence synthesis, you can choose different strategies when identifying and selecting studies to synthesise [24]. For some syntheses, you may include all eligible primary studies, for example, when the qualitative synthesis is coterminous with an effectiveness review. For exploratory or interpretive syntheses, only a sample of the eligible studies may be included. When subsequently assessing relevance, you should reflect on how the sample was located and on the underpinning principles that determined its selection. For example, was the sample selected to explain the attitudes or behaviours of a particular group or to demonstrate the applicability of a theory across groups? In making decisions on how to select the sample of included studies, you may trade-off such factors as the methodological strengths and limitations of available studies (see paper 3 in this series [5]), the relevance of the evidence to the review question and, for instance, the geographical coverage of studies in relation to the review question. For example, if the question is global, review authors may purposively select studies from diverse settings. Conducting an initial knowledge

map of potential studies to identify important contextual factors [29], together with other important elements, may help to design a sampling frame to inform decisions about studies for potential inclusion [8].

Pragmatically, the sampling strategy seeks to optimise the trade-off between the number of available studies that meets the inclusion criteria versus the time available to synthesise studies. A key consideration relates to whether or not a review is required to inform an imminent decision. Where this is the case, the decision maker needs to draw upon the most relevant available evidence and studies may be sampled on that basis. A further consideration is where dissemination bias might limit the identification of relevant evidence to address the question for a specific context. Recent work outlines the conceptual basis for the effects of dissemination bias in qualitative research and its likely extent [30, 31], while the final paper in this series [32] addresses how dissemination bias in qualitative research might impact upon a CERQual assessment.

### Step 3: gather information about relevance in the included studies
Next, you need to gather information from included primary studies to help you to identify similarities between the context of the studies supporting each review finding and the context specified in the review question. Information about context in the primary studies that maps against the context of interest as specified in the review protocol may be reported throughout the primary study. This step is therefore a 'search, find and subsequently extract' exercise. As far as possible, you should ensure that this information is gathered as part of the screening or subsequent data extraction process.

### Step 4: assess the body of data that contributes to each review finding and decide whether you have concerns about relevance
An assessment of relevance is best facilitated by access to subject-specific knowledge either amongst the review authors or from an expert advisory group. To assess the relevance of the data, you need to identify similarities and differences between the context of the studies supporting each review finding and the context specified in the review question. You should assess relevance for each review finding individually and not for the review as a whole. Table 2 provides a non-exhaustive list of potentially important contextual factors when assessing relevance.

We have identified three types of threat to relevance—data that are indirectly relevant to the review question, data that are only partially relevant and data that are of unclear relevance:

**Table 2** Identifying similarities and differences between the context specified in the review question and the context specified in the primary studies contributing to a review finding

Micro-context

*Population characteristics*

• Do particular characteristics related to the population specified in the review question (such as age, gender, socioeconomic status) raise concerns regarding the relevance of the review finding?

• Is the population reported in sufficient detail to make comparisons?

*Characteristics of the setting and place*

• Do particular characteristics related to the setting or place as specified in the review question warrant concerns regarding relevance of the review finding (such as urban versus rural, private versus public, low income versus high income)?

• Are the setting and place reported in sufficient detail to facilitate comparisons?

*Temporal characteristics*

• Are the data likely to be very different from the context specified in the review question because of when these data were collected?

Meso-context

*Intervention characteristics:*

• Do particular characteristics related to the intervention, such as who implemented it and how it was implemented, raise concerns regarding the relevance of the review finding to the review question?

• Is the intervention reported in sufficient detail to make comparisons?

Macro-context

*Policy or political issues, social climate, legislation*

• Do particular socio-political characteristics in the study setting, such as type of government, legality of the intervention, or social and cultural values, raise concerns regarding the relevance of the review finding to the review question?

Cross cutting

*Phenomena of interest*

• Do particular characteristics, or lack of clarity, or lack of reporting concerning the phenomena of interest raise concerns regarding the relevance of the review finding to the question?

1. (Some of) the underlying data are of *indirect relevance*: this assessment is made where a review team is unable to identify studies that fully represent the context of the review, but is able to identify studies that correspond with some factors from the context of the review question but not with others. In other words, one or more aspects of context are substituted with another in these studies. For example, if a synthesis is looking at factors affecting the implementation of public health campaigns for bird flu but finds no, or very little, evidence of direct relevance to bird flu, they may choose to include indirect evidence from studies of swine flu campaigns to address the original synthesis question. This use of indirect evidence is based on the assumption that there are likely to be sufficient common factors between the implementation of the respective public health campaigns. In contrast, other contextual factors such as animal versus bird welfare and containment in a flu epidemic are likely to be very different. Review authors therefore need to be cautious when interpreting indirect evidence where some contextual factors are similar and others are not in order to ensure that the use of indirect evidence is not misleading.

2. (Some of) the underlying data are of *partial relevance*: this is assigned where a part of the context of the larger review question (e.g. a population subgroup) is addressed directly by a review finding but where evidence is lacking for the complete context specified in the review question. For example, a review question seeks to determine Muslim women's experiences of divorce and child custody. A finding that only includes evidence from Jordanian Muslim women may be assessed as "partially relevant" for Muslim women in general. In some circumstances, it may be more appropriate and meaningful to rephrase a finding to incorporate the aspects of context that indicate that the finding is for a particular 'subgroup' in relation to context. Using the same example, the evidence would be reassessed as "directly relevant" for a finding relating to Jordanian Muslim women specifically when this group was pre-specified as a subgroup of interest. Decisions on the level of granularity used when framing the findings depends upon the purpose and intended audience for your specific review.

3. It is unclear whether the underlying data is relevant (*unclear relevance*): this assessment should be reserved for situations where you have identified, typically a priori, important factors that influence the interpretation of review findings but where you are unable to identify those factors from included studies. For example, you have reason to believe that different age groups respond differently to health appointment reminders sent via mobile phone. However, the contributing qualitative studies do not describe the ages of the informants who contributed to specific findings, only supplying a general description of the demographics for all study participants.

Review authors should consider whether any characteristics related to each contextual factor previously identified as important, and reported in primary studies contributing to a finding, are directly, partially or indirectly relevant or of unknown relevance to the context specified in the review question. You may find it helpful to assign each study contributing to a review finding a relevance rating (directly relevant; indirectly relevant; partially relevant; of unclear relevance, with a short explanation). Tables 3, 4, 5 and 6 provide visual examples.

**Table 3** CERQual assessments of relevance in the context of a review finding––examples of no or very minor concerns

Review question: what are the experiences of African women regarding medically assisted birth in public hospitals in African countries?[a]
Finding 1: women feel that they are forced into having a medically assisted birth by medical staff in hospital settings

| Dimensions of context to consider as specified in the question and protocol | Assessment of relevance of each study contributing to the finding mapped against the review question context | | | |
| --- | --- | --- | --- | --- |
| | Direct relevance | Indirect relevance | Partial relevance | Uncertain relevance |
| Time: 2000––present | Study 1 Study 2 Study 3 Study 4 Study 5 Study 6 | | | |
| Place (country): African countries | Study 1 Study 2 Study 3 Study 4 Study 5 Study 6 | | | |
| Phenomenon of interest: African women's experiences of medically assisted birth in public hospitals in African countries | Study 1 Study 2 Study 3 Study 4 Study 5 Study 6 | | | |
| Health system: publicly funded health services | Study 1 Study 2 Study 3 Study 5 Study 6 | – | | Study 4 Health system unclear |
| Population/ perspective: African women's perspectives | Study 1 Study 2 Study 4 Study 5 Study 6 | | Study 3 Participants include non-African women giving birth in African countries | |
| CERQual assessment of relevance[b] | No or very minor concerns about relevance because in one study the setting was unclear. A small number of women whose views contributed to the synthesis were not African but they experienced the same issues giving birth in hospitals in African countries as African women. | | | |

[a]Hypothetical example generated from Bohren et al. [10]
[b]Also see paper 2 in this series on making an overall CERQual assessment [6]

**Table 4** CERQual assessments of relevance in the context of a review finding––examples of minor concerns

Review question: what are the experiences of African women regarding medically assisted birth in public hospitals in African countries?[a]
Finding 1: women feel that they are forced into having a medically assisted birth by medical staff in hospital settings

| Dimensions of context to consider as specified in the question and protocol | Assessment of relevance of each study contributing to the finding mapped against the review question context | | | |
| --- | --- | --- | --- | --- |
| | Direct relevance | Indirect relevance | Partial relevance | Uncertain relevance |
| Time: 2000––present | Study 1 Study 2 Study 3 Study 4 Study 5 Study 6 | | | |
| Place (country): African countries | Study 1 Study 2 Study 3 Study 4 Study 5 Study 6 | | | |
| Phenomenon of interest: African women's experiences of medically assisted birth in public hospitals in African countries | Study 1 Study 2 Study 3 Study 4 Study 5 Study 6 | | | |
| Health system: publicly funded health services | Study 1 Study 2 Study 5 | | Study 3 Study 4 Health system includes public/ private mix | Study 6 Health system unclear |
| Population/ perspective: African women's perspectives | Study 1 Study 2 Study 3 Study 4 Study 5 Study 6 | | Study 3 Participants include non-African women giving birth in African countries | |
| CERQual assessment of relevance[b] | Minor concerns about relevance because in three studies the health systems within which women were treated overlapped, but were not completely congruent, with the context of the synthesis question, or the health system was not reported. A small number of women whose views contributed to the synthesis were not African but they experienced the same issues giving birth in hospitals in African countries as African women. | | | |

[a]Hypothetical example generated from Bohren et al. [10]
[b]Also see paper 2 in this series on making an overall CERQual assessment [6]

You might have fewer concerns about relevance where the contexts of the studies contributing data to a finding match the context of the review question or a pre-identified subgroup. You are not, however, seeking a perfect fit between the included studies and the context of the review question. As with the other CERQual components, the emphasis is on characteristics that trigger significant concerns (see paper 2 in this series on making an overall CERQual assessment [6]). In many cases, you will assess the relevance of review findings that are

supported by studies that are not derived from the exact setting for which the decision is intended. For example, even if one or more individual studies derive from the same country as the decision-making context, you cannot consider these studies to be fully representative of that country. Where multiple studies exist, derived directly from the decision-making context, these studies may represent different time periods within different

**Table 5** CERQual assessments of relevance in the context of a review finding––examples of moderate concerns

Review question: what are the experiences of African women regarding medically assisted birth in public hospitals in African countries?[a]
Finding 1: women feel that they are forced into having a medically assisted birth by medical staff in hospital settings

| Dimensions of context to consider as specified in the question and protocol | Assessment of relevance of each study contributing to the finding mapped against the review question context | | | |
| --- | --- | --- | --- | --- |
| | Direct relevance | Indirect relevance | Partial relevance | Uncertain relevance |
| Time: 2000––present | Study 1 Study 2 Study 3 Study 4 | | Study 5 Study 6 | |
| Place (country): African countries | Study 1 Study 2 Study 3 Study 4 Study 5 Study 6 | | | |
| Phenomenon of interest: African women's experiences of medically assisted birth in public hospitals in African countries | Study 1 Study 3 | | Study 2 Study 4 Study 5 Study 6 | |
| Health system: publicly funded health services | Study 2 | Study 1 Health system private | Study 3 Study 5 Study 6 Health system includes public/ private mix | Study 4 Health system unclear |
| Population/ perspective: African women's perspectives | Study 1 Study 2 Study 3 Study 4 Study 5 Study 6 | | Study 3 Participants include non-African women giving birth in African countries | |
| CERQual assessment of relevance[b] | Moderate concerns about relevance because three studies focussed on birth in general and attitudes to medically assisted birth whether women had as assisted birth or not. The health systems within which women were treated in five contributing studies overlapped with or varied from the context of the synthesis question. A small number of women whose views contributed to the synthesis were not African but they experienced the same issues giving birth in hospitals in African countries as African women. Two studies overlapped with the time period in question and included women whose experiences predated 2000. | | | |

[a]Hypothetical example generated from Bohren et al. [10]
[b]Also see paper 2 in this series on making an overall CERQual assessment [6]

**Table 6** CERQual assessments of relevance in the context of a review finding––examples of serious concerns

Review question: what are the experiences of African women regarding medically assisted birth in public hospitals in African countries?[a]
Finding 1: women feel that they are forced into having a medically assisted birth by medical staff in hospital settings

| Dimensions of context to consider as specified in the question and protocol | Assessment of relevance of each study contributing to the finding mapped against the review question context | | | |
| --- | --- | --- | --- | --- |
| | Direct relevance | Indirect relevance | Partial relevance | Uncertain relevance |
| Time: 2000––present | Study 1 Study 2 | | Study 3 Study 4 | Study 5 Study 6 |
| Place (country): African countries | Study 1 Study 2 Study 3 Study 4 Study 5 Study 6 | | | |
| Phenomenon of interest: African women's experiences of medically assisted birth in public hospitals in African countries | | | Study 1 Study 2 Study 3 Study 4 Study 5 Study 6 | |
| Health system: publicly funded health services | Study 2 | Study 1 Study 5 Health system private | Study 3 Study 6 Health system includes public/ private mix | Study 4 Health system unclear |
| Population/perspective: African women's perspectives African women's perspectives | | | Study 1 Participants include non-African women giving birth in African countries | |
| Population/perspective: African women's perspectives Professional perspectives of women's experiences | | Study 2 Study 3 Study 4 Study 5 | | |
| Population/perspective: African women's perspectives Fathers' perspectives of women's experiences | | Study 6 | | |
| CERQual assessment of relevance[b] | Serious concerns about relevance because the health systems within which women were treated in the contributing studies varied from the context of the synthesis question. In four studies the timeframe overlapped or was different. Five studies reported other actors' interpretations of women's experiences. Only one small study included the perspectives of women and some of the women's views contributed to the synthesis were not African but they experienced the same issues giving birth in hospitals in African countries as African women. | | | |

[a]Hypothetical example generated from Bohren et al. [10]
[b]Also see paper 2 in this series on making an overall CERQual assessment [6]

political and economic settings. In general, we would have no concerns regarding relevance for a review finding where we judge there to be no important differences between the context of the data contributing to the review finding and the context specified in the review question.

Our confidence in a review finding may, however, be reduced where the relation between the contexts of the primary studies and that specified in the review question is not apparent. For example, there may be differences in the perspective or population, the phenomenon of interest or intervention, the context or the timeframe. In

addition, in a review where studies with a direct relation are not found, you may include studies that imperfectly represent aspects of the context. For example, you may use studies conducted with immediately pre- or post-adolescents to inform a review question relating to adolescents if you have been unable to find any studies of the adolescents themselves.

The relevance of a review finding is not related to the number of primary studies contributing data to that finding. For instance, where a finding is based on data from a single study that matches the contextual factors identified in the review question, you would not have concerns about relevance. In contrast, you may have concerns about relevance for a review finding derived from multiple studies if the contexts of those studies do not match the context of the review question. How to assess adequacy of data as part of the CERQual approach is addressed elsewhere in this series [4].

### Step 5: make a judgement about the seriousness of your concerns and justify this judgement

Having completed the process described above, you should decide whether any concerns that you have identified should be categorised as:

- No or very minor concerns regarding relevance
- Minor concerns regarding relevance
- Moderate concerns regarding relevance
- Serious concerns regarding relevance

You should begin with the assumption that there are no concerns regarding relevance for the body of data contributing to each review finding. In practice, very minor or minor concerns will usually not lower our confidence in the review finding, while serious concerns will lower our confidence. Moderate concerns may lead us to consider lowering our confidence, as part of our final assessment of all four CERQual components. Where you have concerns about relevance, you should describe these concerns in the CERQual Evidence Profile in sufficient detail to allow users of the review findings to understand the reasons for the assessments made (for more information, see [6]).

### Implications when concerns about relevance are identified

Concerns about relevance may not only impact on our confidence in a review finding but also may point to ways of improving future research. First of all, these concerns may indicate the need for more primary research that specifically addresses the evidence gap. The review team should also consider whether the review should be updated when this new research becomes available.

Secondly, where the evidence does not match the specificity of the review question, you should also consider contextual or conceptual justifications for including wider evidence to address the review question. For example, in the absence of relevant evidence on what works concerning public health risk communication in swine flu, you could consider using indirect evidence from risk communication in bird flu. You need to also consider whether the use of indirect evidence will impact on your assessment of the other CERQual components.

## Conclusions

Assessing the relevance component requires that you consider potentially important contextual factors at an early stage in the review process. Concerns about relevance impact on assessments of confidence in review findings and are therefore integral to the CERQual approach. However, it is also important to remember that relevance is just one component of the CERQual approach. Having concerns about relevance may not necessarily lead to a downgrading of overall confidence in a review finding as it will be assessed alongside the other three CERQual components.

This paper describes the current thinking of CERQual developers in order to prompt review authors and others to consider issues relating to relevance of findings from qualitative evidence syntheses. However, the CERQual approach in general, and the relevance component in particular, continues to evolve and, in turn, to be informed by ongoing work on the application of research findings more generally.

### Open peer review

Peer review reports for this article are available in Additional file 3.

### Additional files

> **Additional file 1:** Key definitions relevant to CERQual. (PDF 619 kb)
> **Additional file 2:** External relevance of data. (PDF 643 kb)
> **Additional file 3:** Open peer review reports. (PDF 133 kb)

### Acknowledgements

Our thanks for their feedback to those who participated in the GRADE-CERQual Project Group meetings in January 2014 or June 2015 or gave comments to the paper: Elie Akl, Heather Ames, Zhenggang Bai, Rigmor Berg, Karen Daniels, Hans de Beer, Kenny Finlayson, Bela Ganatra, Susan Munabi-Babigumira, Andy Oxman, Vicky Pileggi, Kent Ranson, Rebecca Rees, Holger Schünemann, Elham Shakibazadeh, Birte Snilstveit, James Thomas, Hilary Thompson, Judith Thornton and Josh Vogel. Thanks also to Sarah Rosenbaum for developing the figures used in this series of papers and to the members of the GRADE Working Group for their input. The guidance in this paper has been developed in collaboration and agreement with the GRADE Working Group (http://www.gradeworkinggroup.org).

## Funding

This work, including the publication charge for this article, was supported by funding from the Alliance for Health Policy and Systems Research, WHO (www.who.int/alliance-hpsr/en/). Additional funding was provided by the Department of Reproductive Health and Research, WHO (http://www.who.int/reproductivehealth/en/); Norad (Norwegian Agency for Development Cooperation: www.norad.no), the Research Council of Norway (www.forskningsradet.no); and the Cochrane Methods Innovation Fund. SL is supported by funding from the South African Medical Research Council (www.mrc.ac.za). The funders had no role in study design, data collection and analysis, preparation of the manuscript or the decision to publish.

## Availability of data and materials

Additional materials are available on the GRADE-CERQual website (www.cerqual.org)
To join the CERQual Project Group and our mailing list, please visit our website: http://www.cerqual.org/contact/. Developments in CERQual are also made available via our Twitter feed: @CERQualNet.

## About this supplement

This article has been published as part of *Implementation Science* Volume 13 Supplement 1, 2018: Applying GRADE-CERQual to Qualitative Evidence Synthesis Findings. The full contents of the supplement are available online at https://imple-mentationscience.biomedcentral.com/articles/supplements/volume-13-supplement-1.

## Authors' contributions

All authors participated in the conceptual design of the CERQual approach. JN and AB wrote the first draft of the manuscript. All authors contributed to the writing of the manuscript. All authors have read and approved the manuscript.

## Ethics approval and consent to participate

Not applicable. This study did not undertake any formal data collection involving humans or animals.

## Consent for publication

Not applicable

## Competing interests

The authors declare that they have no competing interests.

# 
## Author details

[1]School of Social Sciences, Bangor University, Bangor, UK. [2]School of Health and Related Research (ScHARR), University of Sheffield, Sheffield, UK. [3]Norwegian Institute of Public Health, Oslo, Norway. [4]Health Systems Research Unit, South African Medical Research Council, Cape Town, South Africa. [5]Uni Research Rokkan Centre, Bergen, Norway. [6]Division of Social and Behavioural Sciences, School of Public Health and Family Medicine, University of Cape Town, Cape Town, South Africa. [7]European Centre for Environment and Human Health, University of Exeter Medical School, Exeter, UK. [8]UNDP/UNFPA/UNICEF/WHO/World Bank Special Programme of Research, Development and Research Training in Human Reproduction, Department of Reproductive Health and Research, WHO, Geneva, Switzerland. [9]Department of Health Management and Economics, School of Public Health, Tehran University of Medical Sciences, Tehran, Iran. [10]Information, Evidence and Research Department, Eastern Mediterranean Regional Office, World Health Organization, Cairo, Egypt. [11]Cochrane, Cochrane Central Executive, London, UK. [12]Department of Family Medicine, Pontificia Universidad Catolica de Chile, Santiago de Chile, Chile. [13]University of North Carolina, Chapel Hill, NC, USA.

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
