## [Open peer review reports. (PDF 133 kb) · Implementation Science : IS]

Open Peer Review reports

Original Submission

26 Oct 2016

Submitted

Original Manuscript

16 Mar 2017

Reviewed

Reviewer Report - Caitlin Kennedy

Overall, this is an interesting paper examining the concept of relevance of data in the CERQual approach to qualitative evidence syntheses. It is generally well-written, though seems to engage with the world of evidence syntheses/GRADE more than it does with concepts from the field of qualitative research. I feel that structured approaches to qualitative evidence summaries could be useful in certain situations (and too reductive in others), and the CERQual approach seems reasonable. I feel this paper can be published in the series with consideration of some issues which I've outlined below.

The methods section is extremely short. What "definitions" did the authors search the literature for, and how? What were the qualitative evidence syntheses that the relevance component was tested with? I see only one example in the paper (from Bohren et al.), which is helpful, but I think more detail in the methods section would be helpful.

A key concept in assessments of rigor in qualitative research is transferability. The "relevance" domain seems to be where transferability would be most relevant to the CERQual approach, yet the term is not mentioned in the manuscript. Transferability is not reducible to whether characteristics such as the country or sub-population are similar across studies (which seem to be what this manuscript focuses on); it is often thought about more as the extent to which theory or elements of theory are relevant to other situations, and it something that is judged by the reader, but facilitated by the author providing thick description and appropriate contextual details. I realize that CERQual is not intended to assess rigor of individual studies, but the overall concept of transferability still seems quite relevant and I'm surprised the authors do not engage with it in the manuscript.

Similarly, the authors talk about concepts such as "internal validity" which are almost never used by qualitative researchers - they imply an objectivist epistemology, where most qualitative researchers follow a constructivist epistemology. This raises the larger point that the current manuscript seems to engage very little with the well-developed body of literature on assessing quality and rigor in qualitative research, which is a concern.

Without having read the other articles in the series (though having read the previous PLoS One paper that outlines the overall CERQual approach), I have some questions about how determining relevance relates to the process of identifying "review findings". Presumably the review findings are most often identified by some synthesis of themes from the studies that have been included in the review. It seems like, if review authors are going to use indirect data (such as the swine flu/bird flu example provided), the reviewers would need to use judgment about which review findings are applicable to their new question (e.g., if the review was about swine flu, a review finding that was related to something about flapping wings or beaks might not be relevant, though a review finding that was related to animal markets might be). Perhaps a little more reflection on how the assessment of relevance relates to the process of identifying review findings could be useful for the end

user.

The authors use concepts like "an initial knowledge map" without defining or referencing them, which may pose a challenge to readers trying to use this manuscript as guidance. When the authors talk about a "sampling strategy", I suggest being more clear that this is referring to how articles are selected for inclusion in the evidence review. (I don't think many readers will be familiar with calling this process a sampling strategy - they most often think of systematic reviews as creating inclusion and exclusion criteria and then trying to identify all studies that meet these criteria, rather than a sample of them.)

Is there any guidance for how review authors should judge concerns about relevance according to the 4 categories listed (no or very minor, minor, moderate, serious)? It seems without some guidance, different review authors could have a hard time figuring out how to categorize concerns.

The article vacillates between addressing the reader as "you" and referring to "review authors" - I suggest making this more consistent.

Minor editorial comments: The abstract and conclusions talk about this paper being "state of the art". I find such terms a bit hyperbolic and self-promoting and not accurate for the life of the paper (the paper will age, as all papers do, and eventually will not be state of the art - a fact recognized two sentences later in the statement: "We expect the CERQual approach, and its individual components, to develop further as our experiences with the practical implementation of the approach increase"). It also should really be for readers to identify what is state of the art. I'd suggest removing this.

There are some typos (e.g., extra/missing parentheses in a couple of places, misspelled words like "threat" instead of "threats", inconsistent tense of verbs in sentences (e.g. heading of Step 4), etc.) that should be corrected. Similarly, there are some references that look like they need editing, and sentences that look like they possibly should include a reference (e.g., "A subsequent paper in the series will seek to address the associated issue of dissemination bias in qualitative research and its likely impact upon a CERQual assessment.")

28 Mar 2017

Reviewed

Reviewer Report - Janet Harris

I enjoyed reading this paper and think it will be a very helpful guide for reviewers.

Abstract: In several parts of the article, you make the important point that the aim of the process is to determine confidence in relation to relevance in terms of the potential goodness of fit between primary studies and the review question. Should this be included more clearly in the abstract?

Defining the focus as current rather than retrospective reviews is helpful.

I'd delete this from opening sentence p. 7 as it distracts from the main point. "as defined from initiation of the review, or emerging iteratively during the review process," Can the sentence "The methodological literature " be rephrased so that it is presented as a rationale?

Step 1 All of the important elements related to clarifying questions and context are here, but they read like a list - which will be fine for experienced reviewers but less helpful for those who are new to reviewing. Can some editing be done to give a clearer structure for the section? It could for example be structured by first addressing where to find important characteristics of context . So paragraph 2 could describe how a priori theory can provide ideas about context; para 3 could then describe micro, meso, macro context.

Specification of sub-group analysis would come after these, the point being that what is identified in terms of context helps to develop the ideas and consider feasibility of sub group analysis. (I k now this is not always the case but it's clear to present something in order and then make the point that it can be iterative).

The paragraph 'Information about the context...' is written in a way that sounds like we are critically appraising rather than constructing a review. Isn't the point that after important elements of context are identified, they need to be placed in several sections of the review to give a coherent strand relating to context throughout?

Step 2: Imminent decisions versus less immediate decision making contexts - very important and useful point!

Step 3: Is quite short and assumes that reviewers will know how to do this. Could a template or example be provided that shows how to make comparisons? And how will you make sure that the process is transparent?

Step 4: Do you mean 'A finding that only includes evidence from Jordanian *Muslim women?'

6 Jul 2017

Author responded

Author comments - Andrew Booth

Reviewer comment	Author response
Reviewer 1	
1. Overall, this is an interesting paper examining the concept of relevance of data in the CERQual approach to qualitative evidence syntheses. It is generally well-written, though seems to engage with the world of evidence syntheses/GRADE more than it does with concepts from the field of qualitative research. I feel that structured approaches to qualitative evidence summaries could be useful in certain situations (and too reductive in others), and the CERQual approach seems reasonable. I feel this paper can be published in the series with consideration of some issues which I've outlined below.	Thank you for this positive feedback.
2. The methods section is extremely short. What "definitions" did the authors search the literature for, and how? What were the qualitative evidence syntheses that the relevance component was tested with? I see only one example in the paper (from Bohren et al.), which is	The methods section was deliberately designed to be brief. The details of the methods used are reported in paper 1 in this series.

helpful, but I think more detail in the methods section would be helpful.	
3. A key concept in assessments of rigor in qualitative research is transferability. The "relevance" domain seems to be where transferability would be most relevant to the CERQual approach, yet the term is not mentioned in the manuscript. Transferability is not reducible to whether characteristics such as the country or sub-population are similar across studies (which seem to be what this manuscript focuses on); it is often thought about more as the extent to which theory or elements of theory are relevant to other situations, and it something that is judged by the reader, but facilitated by the author providing thick description and appropriate contextual details. I realize that CERQual is not intended to assess rigor of individual studies, but the overall concept of transferability still seems quite relevant and I'm surprised the authors do not engage with it in the manuscript.	Thank you for highlighting that while we mention generalisability and applicability we do not mention 'transferability'. As we assume these terms to refer to similar concepts, we have added 'transferability' to our explanation in the paper. The reviewer raises questions about both the relevance component of CERQual and transferability, and the assessment of the methodological limitations of studies that contribute to a review finding. Making an assessment of methodological limitations is one of the four CERQual components and is described in detail in another paper in this series. CERQual is designed to facilitate the use of review findings in decision making processes (see paper 1 in this series). 'Relevance' is the only component that links back to the review question. To apply CERQual, and the relevance component specifically, the review author needs to have sufficient detail about 'context' (in its broadest sense) in the review question, objectives, inclusion, exclusion criteria etc. However, the issue of overall 'confidence' or 'transferability / applicability / generalisability' is captured by the overall CERQual assessment, which includes an assessment of methodological limitations of included studies, the relevance of the contexts of included studies to the review question, and the adequacy and coherence of findings. This difference is noted in this paper and paper 2 in the series discusses the process of making an overall judgement of 'confidence' in findings. The revised text in the manuscript now reads as follows: "CERQual focuses on the assessment of the internal relevance of the body of evidence from included studies contributing to a review finding, as mapped against the context of the review question. This assessment of relevance is not intended to make externally referent claims regarding the transferability, generalisability or applicability (terms that we take to mean the same) of a review finding. Wider external relevance (a concept that maps onto external validity – a term also commonly used by review authors conducting quantitative systematic reviews of the effectiveness of interventions) of a review finding is addressed in part by the overall CERQual assessment.

	This overall assessment, based on judgements for all four CERQual components, seeks to establish the extent to which a synthesis finding is a reasonable representation of the phenomenon of interest [1, 17]. An overall CERQual assessment communicates the extent to which the synthesis finding is likely to be substantially different from the phenomenon of interest, as defined in the review question. For completeness, an explanation of external relevance can be found in Additional file 1.”
4. Similarly, the authors talk about concepts such as "internal validity" which are almost never used by qualitative researchers - they imply an objectivist epistemology, where most qualitative researchers follow a constructivist epistemology. This raises the larger point that the current manuscript seems to engage very little with the well-developed body of literature on assessing quality and rigor in qualitative research, which is a concern.	In line with previous feedback received when publishing papers on CERQual, where possible the concept or process for CERQual or a qualitative evidence synthesis has been mapped onto the broadly equivalent concept or process in GRADE or a quantitative systematic review. The paper has thus been written to speak to systematic reviewers generally and not just review authors who undertake qualitative evidence syntheses. We do however agree with your point and have added a further sentence to speak to the qualitative systematic reviewer and to more clearly explain that internal relevance maps onto 'internal validity' and external relevance maps onto 'external validity'. There is a separate paper in the series on making assessments of methodological limitations of qualitative studies including in a qualitative evidence synthesis.
5. Without having read the other articles in the series (though having read the previous PLoS One paper that outlines the overall CERQual approach), I have some questions about how determining relevance relates to the process of identifying "review findings". Presumably the review findings are most often identified by some synthesis of themes from the studies that have been included in the review. It seems like, if review authors are going to use indirect data (such as the swine flu/bird flu example provided), the reviewers would need to use judgment about which review findings are applicable to their new question (e.g., if the review was about swine	Yes, correct. There is a separate paper in the series that covers the process of reporting findings (in a summary of findings table) that can be used irrespective of the method of qualitative evidence synthesis used to generate the 'themes' or lines or argument etc. We also note in the series that the process of applying CERQual is iterative, and may help review authors to think through the best way of formulating in each review. We have now added a note of caution for review authors to think carefully when using indirect evidence to develop review findings.

flu, a review finding that was related to something about flapping wings or beaks might not be relevant, though a review finding that was related to animal markets might be). Perhaps a little more reflection on how the assessment of relevance relates to the process of identifying review findings could be useful for the end user.	
6. The authors use concepts like "an initial knowledge map" without defining or referencing them, which may pose a challenge to readers trying to use this manuscript as guidance. When the authors talk about a "sampling strategy", I suggest being more clear that this is referring to how articles are selected for inclusion in the evidence review. (I don't think many readers will be familiar with calling this process a sampling strategy - they most often think of systematic reviews as creating inclusion and exclusion criteria and then trying to identify all studies that meet these criteria, rather than a sample of them.)	This is a good point. We have added the appropriate citation for 'knowledge maps'.
7. Is there any guidance for how review authors should judge concerns about relevance according to the 4 categories listed (no or very minor, minor, moderate, serious)? It seems without some guidance, different review authors could have a hard time figuring out how to categorize concerns.	Thank you for raising this. We have constructed four further tables that provide examples of this (Tables 3, 4, 5 and 6).
8. The article vacillates between addressing the reader as "you" and referring to "review authors" - I suggest making this more consistent.	Thank you for pointing this out. The manuscript has been edited for consistency.
9. Minor editorial comments: The abstract and conclusions talk about this paper being "state of the art". I find such terms a bit hyperbolic and self-promoting and not accurate for the life of the paper (the paper will age, as all papers do, and eventually will not be state of the art - a fact	Agree – have removed 'state of the art'.

recognized two sentences later in the statement: "We expect the CERQual approach, and its individual components, to develop further as our experiences with the practical implementation of the approach increase"). It also should really be for readers to identify what is state of the art. I'd suggest removing this.	
10. There are some typos (e.g., extra/missing parentheses in a couple of places, misspelled words like "threat" instead of "threats", inconsistent tense of verbs in sentences (e.g. heading of Step 4), etc.) that should be corrected. Similarly, there are some references that look like they need editing, and sentences that look like they possibly should include a reference (e.g., "A subsequent paper in the series will seek to address the associated issue of dissemination bias in qualitative research and its likely impact upon a CERQual assessment.")	The manuscript has been edited to remove typos.
Reviewer 2	
1. I enjoyed reading this paper and think it will be a very helpful guide for reviewers.	Thank you for your positive comment.
2. Abstract: In several parts of the article, you make the important point that the aim of the process is to determine confidence in relation to relevance in terms of the potential goodness of fit between primary studies and the review question. Should this be included more clearly in the abstract?	The abstract currently covers this important issue in the following sentence: 'When applying CERQual, we define relevance as the extent to which the body of data from the primary studies supporting a review finding is applicable to the context (perspective or population, phenomenon of interest, setting) specified in the review question'.
3. Defining the focus as current rather than retrospective reviews is helpful.	Yes agree – thank you.
4. I'd delete this from opening sentence p. 7 as it distracts from the main point. "as defined from initiation of the review, or emerging iteratively during the review process," Can the sentence "The methodological literature " be rephrased so that it is presented as a rationale?	We would prefer not to delete the opening sentence as it makes the important point the QES questions can be developed a priori or during the review. We are confused by the request to rephrase the sentence commencing 'The methodological literature' as a rationale. This sentence is a statement of fact. As previously requested we have however added an

		additional sentence to explain how 'internal validity' maps onto internal relevance.
5.	Step 1 All of the important elements related to clarifying questions and context are here, but they read like a list - which will be fine for experienced reviewers but less helpful for those who are new to reviewing. Can some editing be done to give a clearer structure for the section? It could for example be structured by first addressing where to find important characteristics of context . So paragraph 2 could describe how a priori theory can provide ideas about context; para 3 could then describe micro, meso, macro context. Specification of sub-group analysis would come after these, the point being that what is identified in terms of context helps to develop the ideas and consider feasibility of sub group analysis. (I know this is not always the case but it's clear to present something in order and then make the point that it can be iterative).	This reorganisation of text is sensible – thank you.
6.	The paragraph 'Information about the context...' is written in a way that sounds like we are critically appraising rather than constructing a review. Isn't the point that after important elements of context are identified, they need to be placed in several sections of the review to give a coherent strand relating to context throughout?	The purpose of this section is to signpost the review author as to where they can find information about context in the primary study. There is no mention of critical appraisal and this confusion has never come up when are facilitating CERQual workshops.
7.	Step 2: Imminent decisions versus less immediate decision making contexts - very important and useful point!	Thank you.
8.	Step 3: Is quite short and assumes that reviewers will know how to do this. Could a template or example be provided that shows how to make comparisons? And how will you make sure that the process is transparent?	We have added Tables 3, 4, 5 and 6 to more clearly illustrate this process.

9. Step 4: Do you mean 'A finding that only includes evidence from Jordanian *Muslim women?	Yes – thank you for pointing this out - we have clarified this in the text.
---	---

Resubmission

6 Jul 2017 Submitted Manuscript version 2

9 Sept 2017 Author responded Author comments - Andrew Booth

General comments from the series editor	Author responses and changes made
Thanks for providing more methodological detail in the overview and subsequent papers. There are still some areas where it would be better if you could provide further details to reflect the amount of international developmental work undertaken e.g. databases searched, timeframes, how literature reviewed etc.	We have added further detail to the overall methods description in paper 1 of the series. Specifically, we have:  - Included the years during which we ran workshops and seminars to obtain feedback on CERQual, and the numbers of workshops and presentations undertaken - Specified the period during which small group feedback sessions were run - Specified the number of CERQual users and Project Group members interviewed In the component papers (papers 3-6), we have noted that the literature searches that we undertook were informal in nature, as follows (example from paper 5): “When developing CERQual’s adequacy component, we undertook informal searches of the literature, including Google and Google Scholar, for definitions and discussion papers related to the concept of adequacy and to related concepts such as data quantity, sample size and data saturation.” We have also elaborated on the methods used to develop the content of paper 7 – please see below.
Ethics statements. Papers state that no humans were involved. Suggest amending to reflect consensus approach, interviews and questionnaires undertaken.	As we did not undertake formal data collection with people – all data collection was informal, in the context of training workshops, presentations and assessments of use of the approach, we have changed the ethics approval and consent to participate statements to the following: “Not applicable. This study did not undertake any formal data collection involving humans or animals.”
Titles and papers could reflect paper nth of # part in a series.	We have changed all titles to the following format, as agreed earlier (example from paper 1):

	'Applying GRADE-CERQual to qualitative evidence synthesis findings – paper 1 of 7: Introduction to the series'
State of the art has been removed from paper 6 but not all of the other papers in the series.	'State of the art' has been removed from all papers in the series.
The new figure outlining the process is a good addition. As a reader I would have found it easier to read papers 3-6/7 before reading paper 2.	As discussed by email with Liz Glidewell, we had a very long debate within the group about this and concluded that there is no perfect order because paper 2 (overall assessment) and papers 3-6 (components) need to be seen together. We placed 'overall assessment' before the component papers as we felt that readers needed to understand what they were working towards before understanding each component. We feel that it would be best to keep the order as it is, but have made the following changes to assist readers: Papers 2, 3, 4, 5 and 6: We have inserted text along the lines of the following (example from paper 2 (p6): 'These component papers are closely related to this paper on making an overall CERQual assessment of confidence and creating a Summary of Qualitative Findings table. We have placed this paper before the four CERQual component papers as we think that it will be helpful for readers to understand how the component assessments will be used before discussing the details of how to apply each component.' Papers 2, 3, 4, 5 and 6: We have included in each paper an additional table that brings together all of the key definitions from each of the papers.
Do you still want to publish paper 7 as a standalone or incorporate it into the overview along with the other ongoing research?	Yes, we feel that it works best as a standalone paper.
Would the figure in the introduction outlining the process work better across all papers in the series as it contains more information than the figure just outlining the 4 and probable 5 th component?	Thanks for this very helpful suggestion which we have implemented across all of the papers.
1. Introduction	
The lack of such methods constrains the use of...suggest reframing to "methods may constrain".	Change made

“The CERQual approach is intended to be applied to well conducted syntheses.” Could this be confusing to those applying the four components? Isn't CERQual designed to provide evidence of confidence in a well conducted syntheses?	We have not found this to be confusing in our interactions with users of CERQual. We feel that there would be little point in applying CERQual to a synthesis that has been poorly conducted as the findings of such a synthesis are unlikely to be reliable and the synthesis is unlikely report transparently the methods used or to include sufficient information on the primary studies to allow a CERQual assessment to be undertaken. We take the same approach in relation to GRADE for effectiveness, for the same reasons. The problem is sometimes colloquially called 'garbage in-garbage out'!
The section “Applying CERQual across types of qualitative data and syntheses methods”. Would this be better placed after outlining how CERQual was developed?	We agree and have moved this section.
“supported other teams”. Can you say any more about the scale or settings involved?	We have provided more detail as follows: “Thirdly, we applied the CERQual approach within diverse qualitative evidence syntheses in the areas of health and social care [6-8, 26-33] and also supported other teams in using CERQual by providing guidance through face-to-face or virtual training meetings and commenting on draft Summaries of Qualitative Findings tables. At least ten syntheses were supported in this way (for example, [34, 35]).”
Can you provided further detail about the questionnaire and qualitative interviews?	We have now provided further detail in the text and added an additional file listing the questions covered. The revised text reads as follows: “We then gathered structured feedback from early users of CERQual through an online feedback form that was made available to all CERQual users and through short individual discussions with six members of review teams and two members of the CERQual Project Group. The questions included in the online feedback form and individual discussions are available in Additional File X.”
Summarise important areas for methodological research from table 4 in text for the readers ease?	We have revised the text as follows: “Table 4 identifies several important areas for further methodological research, including how to apply CERQual in syntheses that include qualitative and quantitative data; how best to present CERQual assessments together with other kinds of evidence; ways of applying CERQual to syntheses of sources that have not used formal qualitative research procedures; and whether CERQual requires adaptation for application to more interpretive synthesis outputs, such as logic models.”
2. Making an overall assessment and summary of qualitative findings	

Should the paragraph describing the four levels and rating down on p12 be moved to p10 under the 4 bulleted levels of concern?	This change has been made.
Place the text relating to variation in assessors after the text outlining who should undertake an assessment?	This change has been made.
Table 5. typo in component t missing. Should you advise assessors to report how they've handled variation in levels of concern?	This typo has been corrected.
3. Methodological limitations – problems design or conduct of primary studies	
Consider adding a brief description of the Evidence Profile to p12.	Ok. We have now added the following parentheses describing the evidence profile on page 12 following the sentence: "Where you have concerns about methodological limitations, describe these concerns in the CERQual Evidence Profile in sufficient detail to allow users of the review findings to understand the reasons for the assessments made (The Evidence Profile presents each review finding along with an explanation of its CERQual assessment)"
Link in text to table 2?	We have now added the following on page 9: "See Table 2 for an outline of areas where further work is needed with respect to critical appraisal tools for qualitative research."
4. Coherence – How well finding supported by body of evidence 3500 3429	
Consider adding a brief description of the Evidence Profile to p13.	We have added a brief description of the evidence profile on page 12: "Where you have concerns about coherence, you should describe these concerns in the CERQual Evidence Profile in sufficient detail to allow users of the review findings to understand the reasons for the assessments made. The Evidence Profile presents each review finding along with the assessments for each CERQual component, the overall CERQual assessment for that finding and an explanation of this

	overall assessment. For more information, see the second paper in this series [19].”
5. Adequacy of data – degree of richness and quantity of data 3500 2507	
Consider contacting authors for further information as in other assessments?	We have added the following information to lines 204-205: “An overview of the number of studies from which this data originated, and where possible, the number of participants or observations. Information about the number of participants or observations supporting each finding may be difficult to gain from the individual studies. While most studies describe the number of participants they included in their study overall or give some indication of the extent of their observations, they may be less clear about how well represented participants are in different themes and categories. You can contact study authors for additional information, but they may not be able to readily provide this level of detail. In these cases, this lack of information should be noted, and your assessment of data adequacy will have to be made based on the information available.”
The sentence “For a description on descriptive and explanatory findings...” isn’t embedded.	We have moved this sentence to lines 232-233.
Consider adding a brief description of the Evidence Profile to p12.	We have added the following information to lines 277-279: The Evidence Profile presents each review finding along with the assessments for each CERQual component, the overall CERQual assessment for that finding and an explanation of this overall assessment.
6. Relevance – extent applicable to context (perspective or population, phenomenon of interest, setting) of review question 3500 3551	
I found a lot of the text more relevant to conducting a review than the CERQual	Relevance is the only CERQual component that links directly to the review question. All the issues raised

assessment e.g. using theories and frameworks, how and when the review question should be developed, the pre-specification of sub-groups, strategies for identifying and selecting studies, trade-offs in searching.	by the Editor need to be taken into consideration at the review design stage. We make this clear in the manuscript. See P6: 'Relevance is the CERQual component that is anchored to the context specified in the review question. How the review question and objectives are expressed, how a priori subgroup analyses are specified, and how theoretical considerations inform the review design are therefore critical to making an assessment of relevance when applying CERQual.' See page 11: 'When assessing relevance, you should reflect on how the sample was located and on the underpinning principles that determined its selection....'
Word missing p13 "You should if possible, that this"	Sincere apologies, this typo was corrected previously but the corrected draft was not uploaded last time.
Is it possible to comment on how the levels of concern map onto the different threats to relevance 'partial', 'indirect' and 'unclear'?	Tables 3, 4, 5 and 6 provide visual examples. Sincere apologies, these tables may not have been uploaded in error last time.
7. Dissemination bias – selective dissemination of studies or findings 2000 2455	
Methodological details e.g. 'consulting relevant literature' and 'additional empirical work'	We have added further detail as follows: Abstract: "We developed this paper by gathering feedback from relevant research communities, searching MEDLINE and Web of Science to identify and characterize the existing literature discussing or assessing dissemination bias in qualitative research and its wider implications, developing consensus through project group meetings, and conducting an online survey of on the extent, awareness and perceptions of dissemination bias in qualitative research." Main text:

		“We used a pragmatic approach to develop our ideas on dissemination bias by consulting the literature on this topic, including searching MEDLINE and Web of Science to identify and characterize the existing literature discussing or assessing dissemination bias in qualitative research and its wider implications [3]; talking to experts in dissemination bias and qualitative evidence synthesis in a number of workshops; and developing consensus through multiple face-to-face CERQual Project Group meetings and teleconferences. We also undertook an online survey of researchers, journal editors and peer reviewers within the qualitative research domain on the extent, awareness and perceptions of dissemination bias in qualitative research [4].”
--	--	---

Resubmission 2

9 Sept 2017 Submitted Manuscript version 3

Publishing

17 Oct 2017 Editorially accepted

How does Open Peer Review work?

Open peer review is a system where authors know who the reviewers are, and the reviewers know who the authors are. If the manuscript is accepted, the named reviewer reports are published alongside the article. Pre-publication versions of the article and author comments to reviewers are available by contacting info@biomedcentral.com. All previous versions of the manuscript and all author responses to the reviewers are also available.

You can find further information about the peer review system here.